# Controlling the sign of optical forces using metaoptics

Adeel Afridi [1,2], Bruno Melo[1,2], Nadine Meyer [1,2] ✉ & Romain Quidant [1,2] ✉

Precise manipulation of small objects using light holds transformative potential across diverse fields. While research in optical trapping and manipulation predominantly relies on the attraction of solid matter to light intensity maxima, here we demonstrate that meta-optics enables a departure from this accepted behavior. Specifically, we present deterministic control over the sign of optical forces exerted on a metasurface integrated on a suspended silicon nanomembrane. By tailoring the geometry of the constituent meta-atoms, we engineer the coherent superposition of their multipolar modes, and consequently, the net optical force experienced by the metasurface within a phase-controlled optical standing wave. In excellent agreement with 3D numerical simulations, we experimentally realize both attractive and repulsive forces on distinct metasurface designs, directly mirroring the behavior of two-level systems interacting with optical fields. This work establishes a versatile platform for the optical control of nanoscale mechanical systems, opening alternative avenues for both fundamental research and engineering.

The interaction of light with matter involves the fundamental exchange of photon momentum, resulting in radiation pressure. This phenomenon, predicted by Maxwell's electromagnetic theory, was experimentally confirmed in 1901 by Lebedev[1], and Nichols and Hull[2], followed by the observation of photon recoil by Poynting and Barlow[3]. Although radiation pressure under terrestrial conditions is usually too small to be noticed, it plays an important role in the formation of stars[4] and in the dynamics of spacecrafts in outer space[5,6], where the radiation pressure is the main force next to gravity. Nowadays, radiation pressure is even harnessed on demand, for instance in photonic sails to propel ultralight satellites through space[7–9] or to control optomechanical systems across a wide range of sizes[10–12].

However, light-matter interaction is not limited to repulsive forces. Ashkin's seminal work revealed that polarizable objects placed in an inhomogeneous light field experience a force that attracts them towards the highest intensity[13–15]. This discovery laid the foundation for the field of optical tweezers, enabling the precise control and manipulation of microscopic particles. Nowadays, optical tweezers are widely used in cell-biology[16–18], climate research with aerosols[19], quantum optics and quantum simulations with ultracold atoms and molecules[20,21], and levitation optomechanics[22,23], among others. Along

the same lines, specimens with lower refractive index than the environment are attracted to intensity minima[24–26], especially important for light sensitive biological samples. Further experimental exploration revealed surprising effects, such as pulling forces from engineered unfocused light beams, coined tractor beams (see discussion in[27–30]) or non-reciprocal optical binding[31,32]. Recently, emerging metamaterials have expanded the optical manipulation toolkit beyond conventional beam engineering, enabling direct control over the object subjected to optical forces[33]. This advancement facilitates the generation of lateral and transversal optical forces[34–36] which are employed in the two-dimensional steering of metavehicles for the targeted delivery of unicellular organisms.

The optical forces experienced by a specimen exposed to a light intensity gradient are hereby fundamentally governed by its polarizability, offering additional control through its geometrical and structural properties. This was highlighted in 1977 when Askhin and Diedzic observed enhanced radiation pressure along the optical axis on levitated oil drops from a probe beam spectrally matching their Mie resonances [37]. Similar resonant effects were also studied in plasmonic nanoparticles[38–40] and nanodiamonds hosting multiple nitrogen-vacancies[41]. More recently, drawing an analogy to two-level atoms[42],

[1]Nanophotonic Systems Laboratory, Department of Mechanical and Process Engineering, ETH Zurich, 8092 Zurich, Switzerland. [2]Quantum Center, ETH Zurich, 8083 Zurich, Switzerland. ✉e-mail: nmeyer@ethz.ch; rquidant@ethz.ch

we proposed to exploit electromagnetic Mie resonances in high-permittivity meta-atoms for optical trapping at intensity minima[43], by exploiting engineered multipolar resonances[44,45], an effect that has been theoretically anticipated in earlier works[46,47]. Furthermore, unconventional switchable optical forces were reported in nanoparticles made of temperature-sensitive phase change materials [48].

In this work, we demonstrate full control over longitudinal optical forces acting on a metasurface[49–52]. Through precise meta-atom design, we experimentally achieve a controllable reversal of the optical forces, from repulsive to attractive, in excellent agreement with 3D multi-physics simulations. Leveraging solely geometry and intrinsic material properties, our method generates these resonant forces without complex beam engineering, offering possibilities for scalable light-based manipulation of matter.

## Results

### System description

The studied configuration features a free-standing metasurface placed in an optical standing wave, as illustrated in Fig. 1a. The metasurface design consists of a periodic array of identical silicon discs, each with radius $R$, separated by short connectors of length $S$ (see Fig. 1b). To minimize mechanical stress and allow free motion, the metasurface is suspended from a frame using undulated tethers[53]. Given that a single optical beam carries significant photon momentum along its propagation direction, it cannot generate a pulling force[27,54]. Hence, we opted for a standing wave field formed by two counter-propagating Gaussian beams. This configuration cancels the scattering force,

thereby ensuring that the gradient force dominates. Within the Gaussian beam approximation, the gradient force experienced by the metasurface is given by $\mathbf{F}(z) \approx \alpha_{\text{eff}} \nabla I(z)$ where $\alpha_{\text{eff}}$ represents the real part of the effective polarizability, and $\nabla I(z)$ the intensity gradient at the metasurface position $z_0$. This is intuitive because higher-order multipoles correspond to higher order derivatives, which change sign for a linearly polarized field $\mathbf{E} \propto \hat{\mathbf{y}} E_0 \cos(kz)$ at each order (see Supplementary Information Sec.2). The effective polarizability $\alpha_{\text{eff}}$ is primarily determined by the magnetic and electric dipolar, quadrupolar and octupolar modes[27,43] supported by each meta-atom.

Using this configuration, we aim to demonstrate accurate control over both the magnitude and direction of the total force by tuning geometrical parameters to engineer the coherent superposition of Mie multipolar modes [27]. Specifically, we leverage the dependence of the scattering radiation pattern of silicon discs, and hence the net optical force $F_0$ they experience, on the ratio of their radius $R$ to the laser wavelength $\lambda$[43]. By varying disc radii $R$ and separations $S$, we explore a wide range of force amplitudes and directions. Depending on the optical wavelength, each metasurface acts as a high- or low-field seeker attracted to intensity maxima or minima, in full analogy to two-level systems[42]. Note that force amplitude and direction is polarization independent due to the symmetry of the meta-atom array (see Supplementary Information Sec.3).

### Theoretical model

To estimate the total optical force experienced by the freely moving metasurface, we exploit its resonant mechanical mode, which exhibits

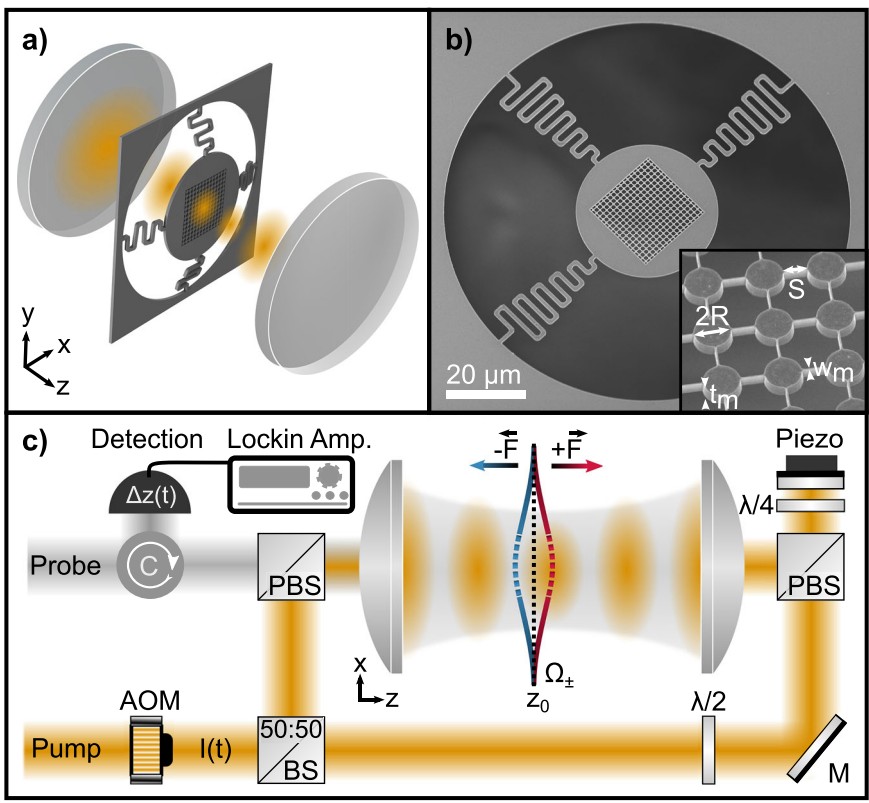

**Fig. 1 | Experimental configuration. a** A silicon membrane, patterned with a metasurface, is suspended within the standing wave created by two phase-stabilized, counterpropagating beams focused by a pair of lenses. Undulated tethers allow the free standing membrane to move along $z$. **b** SEM picture of tethered, patterned membrane of thickness $t_m$ featuring a symmetric disc array with disc radius $R$, separated by connectors of length $S$ and width $w_m$ (see inset). **c** The suspended membrane with mechanical eigenfrequency $\Omega_m = \Omega_\pm$ is positioned at $z_0$ along the standing wave (yellow). The standing wave phase $\phi_0$ is controlled with a

piezo-actuated mirror. The light intensity $I(t)$ is modulated by an acousto-optical modulator (AOM). The force vector $F_+$ ($F_-$) indicates a membrane displacement towards $z > z_0$ ($z < z_0$). The probe beam (gray) is phase modulated by $z(t)$ and used for homodyne detection of the membrane displacement where a lock-in detection extracts both phase $\Theta(\omega)$ and amplitude $\mathcal{A}(\omega)$. The optical setup includes polarizing beam splitters (PBS), beam splitters (BS), mirrors (M), and half ($\lambda/2$) and quarter waveplates ($\lambda/4$).

**Table 1 | Parameters for membrane $M_+$ and $M_-$ used in Fig. 2 and Fig. 4 with radius $R$, separation $S$, mechanical eigenfrequency $\Omega_\pm$, damping $\Gamma_\pm$ and $\phi_s$**

|       | $R$ [nm] | $S$ [nm] | $\Omega_\pm/(2\pi)$ [kHz] | $\Gamma_\pm/(2\pi)$ [kHz] | $\phi_s$ |
|-------|----------|----------|--------------------------|---------------------------|----------|
| $M_+$ | 345      | 530      | 94.81                    | 10.32                     | $-\pi/2$ |
| $M_-$ | 485      | 430      | 92.55                    | 9.54                      | $+\pi/2$ |

The parameters $\Omega_\pm$ and $\Gamma_\pm$ are obtained from the fit in Fig. 2a.

a frequency-dependent amplitude response to an external driving force[55,56]. We model the system as a driven underdamped harmonic oscillator, characterized by the oscillator's mechanical eigenfrequency $\Omega$, its mass $m$ and the amplitude decay rate $\Gamma$ due to clamping losses and gas collisions at atmospheric pressure. The optical driving force is generated by modulating the intensity of the standing wave pattern $I(t) = I_0(1 - \cos(\omega_{dr}t))/2$, such that a time-dependent optical force $F(t) = F_0(1 - \cos(\omega_{dr}t))/2$ is exerted on the membrane. In this context, the optical driving force $F(t)$ prevails over the thermal driving force (see Supplementary Information Sec.6). The displacement of the oscillator, $z(t) = \mathcal{A}(\omega)\cos(\omega t + \theta(\omega))$, is characterized by its frequency dependent amplitude $\mathcal{A}(\omega) \propto F_0$ and its phase $\theta(\omega)$ across the resonance.

Eventually, the sign of the intensity gradient at the membrane's equilibrium position along the standing wave dictates the direction of the displacement. While the amplitude and phase relationship between $z(t)$ and $F(t)$ are invariant for high- and low-field seekers, their displacement directions are opposite (see Fig. 1c). However, since measurements are referenced to the driving signal of $I(t)$, this results in an effective reversal of the force $F(t)$ and a phase shift in the measured motion $z(t)$ relative to the modulation signal, when comparing high- and low-field seekers at the same position. The measured phase between $I(t)$ and $z(t)$ can thus take on two distinct values, given by:

$$\Theta(\omega) = \theta(\omega) + \phi = \begin{cases} \theta(\omega) + 0 \text{ for } F_+ \\ \theta(\omega) + \pi \text{ for } F_- \end{cases} \quad (1)$$

where we refer to in-phase (out-of-phase) oscillations as positive (negative) force $F_+$ ($F_-$). Note that, depending on the sign of the intensity slope, both high- and low-field seekers can exhibit positive and negative force behavior. To account for this we define $\phi = |\phi_0 + \phi_s|$. The phase term $\phi_s = +\pi/2$ ($\phi_s = -\pi/2$) corresponds here to a low- (high-) field seeker that is repelled (attracted) by high intensity. This consequently leads to $\phi_0 = +\pi/2$ ($\phi_0 = -\pi/2$) for the positive (negative) intensity slope of the standing wave pattern.

**Experimental implementation**

The experimental configuration is illustrated in Fig. 1c. Two counter-propagating, equally $y$-polarized beams at a wavelength $\lambda = 1550$ nm with power $P = 20$ mW are focused by two lenses of numerical aperture NA=0.4 resulting in a beam waist smaller than the meta-surface area. Both beams are phase-stabilized to form an interference pattern along the optical axis $z$ (shown in yellow) where the relative position of intensity maxima and membrane position $z_0$ is controlled with the beams' relative phase $\phi_0$. The optical external driving force is generated by modulating the optical intensity $I(t)$ of this standing wave with an acousto-optic modulator (AOM) at $\omega_{dr}$ and amplitude $I_0$. The metasurface, consisting of an array of identical discs, is patterned on freestanding crystalline silicon membranes of thickness $t_m = 350$ nm[57]. The discs in the array are connected by thin nano-beams of width $w_m = 70$ nm. An additional single cross-polarized beam co-propagates along the optical axis (depicted in gray in Fig. 1c). The backscattered light of this probe beam is phase modulated by the motion of the membrane $z(t)$ and therefore can be used

for optical displacement readout via phase sensitive homodyne detection[58]. By using a lock-in operation, we detect the membrane motion $z(t) = \mathcal{A}(\omega_{dr})\cos(\omega_{dr}t + \theta(\omega_{dr}) + \phi)$ with the relative phase $\theta(\omega_{dr}) + \phi$ between the membrane motion and the reference signal $\propto \cos(\omega_{dr}t)$ (see Supplementary Information Sec.4).

We first focus on two membranes (see Table 1), $M_+$ and $M_-$, which exhibit high- and low-field seeker behavior, respectively. These membranes are positioned at the rising slope of the intensity field ($\phi_0 = \pi/2$). Fig. 2 depicts the motional response of membranes $M_+$ and $M_-$ under the driving force $F(t)$ with varying $\omega_{dr}$ at atmospheric pressure. The amplitude $\mathcal{A}_\pm(\omega_{dr})$ is normalized by the experimental amplitude response of an unstructured flat membrane $\mathcal{A}_{RP}(\Omega)$ acting as a mirror. This mirror membrane experiences radiation pressure force from a single beam $F_{RP} = \frac{P}{c}[2r + a]$[59], where $r$ and $a$ are the reflectivity and absorption of the unstructured flat membrane, $P = 40$ mW the optical power, and $c$ the speed of light (see Supplementary Information Sec.5). Importantly, the single beam reference measurement allows a reliable calibration of the phase $\Theta(\omega)$ due to the defined force direction along the beam propagation axis for an unstructured flat membrane under single beam illumination.

As shown in Fig. 2a, the driven motion exhibits the typical Lorentzian profile of a mechanical resonance, centered at $\Omega_\pm$ (solid lines) and with a linewidth $\Gamma_\pm$ (see Table 1). Figure 2b displays the phase response $\Theta(\omega_{dr})$ of the driven membranes $M_+$ and $M_-$. The dashed lines represent fits to the theory (see Supplementary Information Sec.4). We observe that the membrane $M_+$ (red) oscillates in phase with the reference signal at low $\omega_{dr}$. Upon exceeding its resonance frequency ($\omega > \Omega_+$) it undergoes the expected $\pi$ phase jump. indicating that the oscillator response lags behind the driving force. In contrast, $M_-$ oscillates out of phase ($\Theta = \pi$) already for $\omega < \Omega_-$ and changes to an in-phase oscillation with respect to the reference signal ($\Theta = 0$) across the resonance. The different phase responses, in excellent agreement with the theory, demonstrate that the membranes $M_+$ and $M_-$ experience opposite force signs (see insets in Fig. 2b). Specifically, $M_+$ acts as a high-field seeker, drawn to the intensity maximum, whereas $M_-$ behaves as a low-field seeker, drawn atypically to the intensity minimum, a scenario difficult to achieve with dielectric structures in the dipole regime[60].

To demonstrate complete control over the optical force and its dependence on structural parameters, we fabricate a series of meta-surfaces systematically exploring the $(R, S)$ parameters space where the radius $R$ ranged from 250 to 550 nm and the separation ranged from 430 to 730 nm. For each metasurface, we position the membrane at $\phi_0 = \pi/2$ and modulate the intensity at $\omega_{dr} = \Omega$ for each metasurface.

In Fig. 3a we compare the measured force amplitudes $F_0/(F_{RP}\mathcal{F}) = [\mathcal{A}_\pm(\Omega_\pm)\Gamma_\pm\Omega_\pm]/[\mathcal{A}_{RP}(\Omega)\Gamma_{RP}\Omega_{RP}]$ (circles) with simulation. COMSOL Multiphysics simulation in a scattered-field formulation models a single silicon meta-atom suspended in air within a periodic unit cell. Periodic boundary conditions (PBC) are applied laterally, and perfectly matched layers (PML) along the propagation axis to suppress reflections. A normally incident standing wave provide the excitation, and the optical forces are extracted via Maxwell stress tensor methods[61] using the total electromagnetic field (see Supplementary Information Sec.3). The geometric filling factor $\mathcal{F} = (\pi R^2 + 2Sw_m)/(2R + S)^2$ is equal to the mass ratio of the metasur-face and mirror membrane. The contourplot displays the simulations. Figure 3b and c show line plots for constant separations $S = 430$ nm and 530 nm exhibiting the largest negative force $F_-$ and largest positive force $F_+$ with a magnitude comparable to the radiation pressure $F_{RP}$. The fabrication uncertainty of $\delta R = \pm 10$ nm and $\delta S = \pm 20$ nm is represented by the experimental error bars and the gray-shaded area in the simulation. We observe a broad tunability of the force $F_0$ that is only weakly affected by the separation $S$, with its strongest influence occurring in the intermediate regime ($R \simeq \lambda/3$). We attribute this to the scattered field stemming from the connecting beams near the disc and

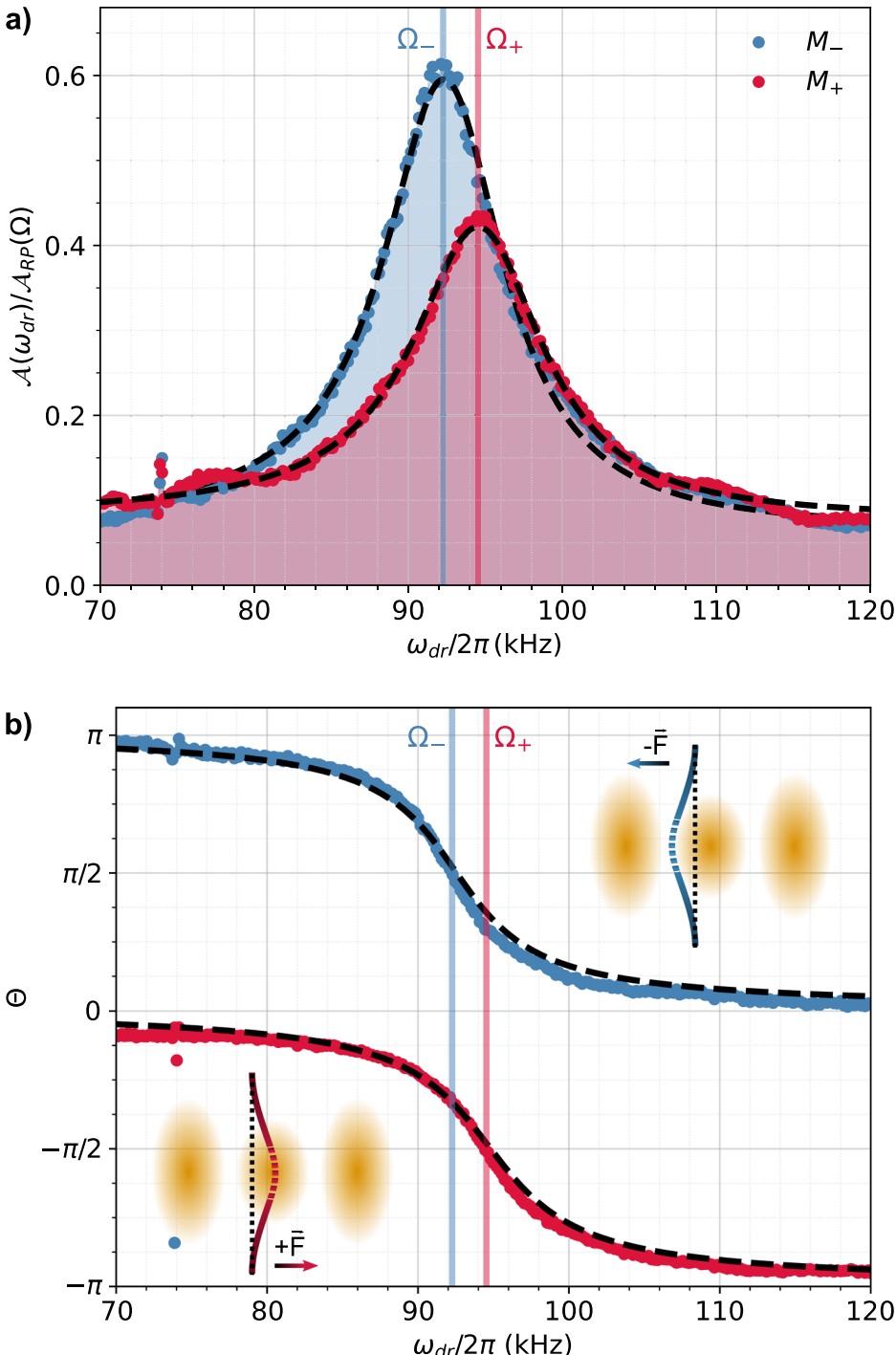

**Fig. 2 | Metasurfaces with positive and negative force response $F_\pm$. a** Normalized driven displacement amplitude $\mathcal{A}(\omega_{dr})/\mathcal{A}_{RP}(\Omega)$ for membranes $M_+$ (red) and $M_-$ (blue) around their respective resonance $(\Omega_+, \Omega_-)/(2\pi) = [94.81, 92.55]$ kHz (solid lines) at atmospheric pressure. **b** Driven phase response $\Theta(\omega_{dr})$ for $M_+$ (red) is in-phase $(|\Theta(\omega_{dr})| \approx 0)$ below the resonance $(\omega < \Omega_+)$ and out-of-phase $(|\Theta(\omega_{dr})| \approx \pi)$ above the resonance $(\omega > \Omega_+)$, indicating a positive force response $F_+$. The $M_-$ membrane (blue) shows a negative force response $F_-$, demonstrating the opposite phase response $(|\Theta(\omega_{dr})| \approx \pi$ for $\omega < \Omega_-$ and $|\Theta(\omega_{dr})| \approx 0$ for $\omega > \Omega_-)$. Dashed lines are fits to the theory and the solid lines highlight the resonance frequency $\Omega_\pm$. The insets indicate the membrane motion for $F_\pm$ towards high or low intensities.

the disc itself. However, at smaller radii, the force amplitude tends to be larger with frequent negative values, whereas at midrange radii, the amplitude is predominantly positive. We find experiment and simulation in good agreement, confirming the tunability of optical forces by structural parameters $R$ and $S$.

As noted above, the sign of the optical force depends on both the membrane's high- or low-field seeking behavior $(\phi_s)$ and its position

within the standing wave, determined by the intensity gradient $(\phi_0)$. To demonstrate the position dependence of the relative phase $\phi$ between optical force $F(t)$ and reference signal, we sweep the standing wave relative to the membrane's position $z_0$ across several wavelengths by tuning the phase $\phi_0$, while simultaneously driving the membrane's motion at $\omega_{dr}/(2\pi) = 85$ kHz. For this analysis, we deploy $M_+$ and $M_-$ (see Table 1). Figure 4a shows the theoretical intensity $I_{SW}/I_0$

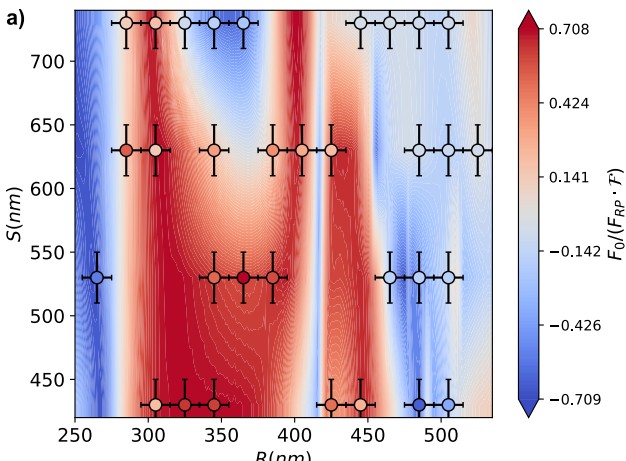

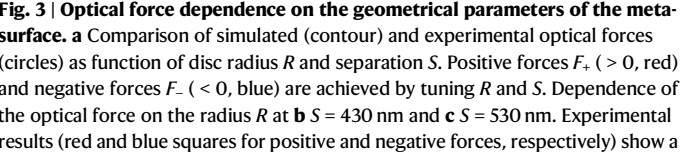

**Fig. 3 | Optical force dependence on the geometrical parameters of the metasurface. a** Comparison of simulated (contour) and experimental optical forces (circles) as function of disc radius $R$ and separation $S$. Positive forces $F_+$ ( > 0, red) and negative forces $F_-$ ( < 0, blue) are achieved by tuning $R$ and $S$. Dependence of the optical force on the radius $R$ at **b** $S = 430$ nm and **c** $S = 530$ nm. Experimental results (red and blue squares for positive and negative forces, respectively) show a

non-trivial dependence on the radius, consistent with COMSOL simulations (solid lines). Both experimental data and simulations are normalized to the radiation pressure force of a mirror corrected by the filling factor $\mathcal{F}$ (see Supplementary Information Sec.5). Error bars on the experimental data represent uncertainties due to fabrication tolerances $\delta R = \pm 10$ nm and $\delta S = \pm 20$ nm. The gray shaded area accounts for the fabrication error in the simulations.

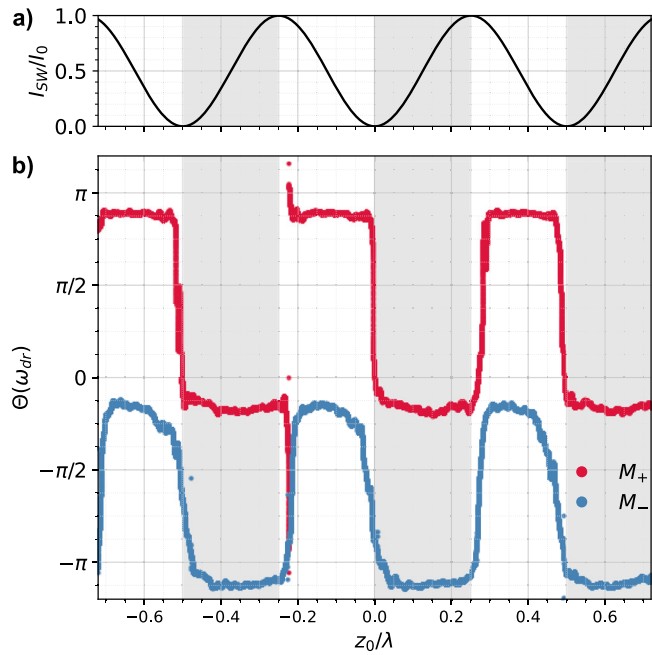

**Fig. 4 | Position dependence of optical forces $F_\pm$. a** Theoretical, normalized intensity distribution $I_{SW}/I_0$ along the optical axis $z$ exhibiting intensity minima and maxima. Gray (white) shaded areas correspond to a positive (negative) intensity slope $dI/dz > 0$ ($dI/dz < 0$). **b** Phase response $\Theta(\omega_{dr})$ of membranes $M_+$ (red) and $M_-$ (blue). Membrane $M_+$ exhibits in-phase motion, acting as a high-field seeker whereas $M_-$ shows out-of-phase motion, behaving as a low-field seeker. Both membranes experience a phase jump of $|\Theta(\omega_{dr})| = \pi$ at points where $dI/dz = 0$. The driving frequency for both membranes is $\omega_{dr} = 2\pi \times 85$ kHz which is below their respective resonance frequencies $\Omega_\pm$.

along the standing wave. The measured lock-in phase $\Theta(\omega_{dr})$ for $M_+$ (red) and $M_-$ (blue) is shown in Fig. 4b, displaying a constant phase difference of $\pi$ between $M_+$ and $M_-$. For a negative intensity slope ($\phi_0 = -\pi/2$), $M_+$ exhibits a phase shift of $\Theta(\omega_{dr}) \approx \pi$, while $M_-$ remains in phase at $\Theta(\omega_{dr}) \approx 0$. The phase shifts reverses sweeping to a positive

intensity gradient ($\phi_0 = +\pi/2$), further confirming the opposite signs of optical forces in these two metasurfaces. The phase discontinuity observed at $z_0/\lambda = -0.2$ arises from phase wrapping in the lock-in detection limited to $\pm\pi$. This implies a sudden phase jump by $2\pi$ when $\Theta(\omega_{dr})$ crosses $\pm\pi$.

## Discussion

In summary, we demonstrate deterministic control over both the amplitude and direction of longitudinal optical forces on suspended high-refractive-index metasurfaces via advanced mode engineering. Most notably, we unveil the counterintuitive phenomenon of low-field-seeking behavior, its attraction to intensity minima, enabled by the interplay of Mie multipolar modes. Importantly, this is a resonant phenomenon, enabling the tuning between attractive and repulsive forces solely through wavelength adjustment. Beyond its fundamental significance, this approach establishes a powerful and versatile platform for optical force engineering, with potential extensions to optical torques[62]. The experimental demonstration of multipolar optical force control holds relevance for diverse applications, including the minimization of photodamage in biological systems[63], resonant nanoparticle sorting[64], optical manipulation[38] for precise nanostructure assembly, steering of metavehicles for drug delivery[34–36], optically reconfigurable photonic devices[57], near-surface trapping[42], and optomechanics in general.

## Methods

### Fabrication of metasurfaces

To fabricate the metasurfaces, we employ top-down electron beam lithography. Commercially available free standing crystalline (100) silicon membranes from Norcada Inc. serve as material substrate. The membrane sample is spin coated with the AR-P 6200.04 positive photo-resist with a thickness of 230 nm followed by baking for 1 minute at 150˚C. Afterwards, electron beam exposure is carried out followed by 90 seconds in the AR 600-546 developer at room temperature. We etch the silicon membrane using HBr chemistry with an inductively coupled plasma etcher. Finally, the photo-resist is stripped off with an oxygen plasma etcher. The patterned metasurface of area $20\,\mu m \times 20\,\mu m$ is placed in the center of a circular membrane of diameter $D = 36\,\mu m$ which is connected by undulated tethers to the substrate frame.

## Data availability

The data generated in this study have been deposited in the ETH Zürich Research Collection at[65].

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

## Acknowledgements

This research was supported by the European Research Council (ERC) through grant Q-Xtreme ERC 2020-SyG (grant agreement number 951234). We acknowledge valuable discussions with the Q-Xtreme synergy consortium.

## Author contributions

A.A. performed the numerical simulations, fabricated the device, performed the measurements and analyzed the data. A.A. and B.M. designed and implemented the optical setup. N.M developed the theoretical expressions. N.M. and R.Q. conceptualized the experiments. All authors discussed the results and contributed to writing the manuscript.

## Funding

## Competing interests

The authors declare no competing interests.
