## [Transparent Peer Review file · Nature Communications]

Controlling the sign of optical forces using metaoptics

Corresponding Author: Dr Nadine Meyer

Version 0:

Reviewer comments:

Reviewer #1

(Remarks to the Author)

The manuscript demonstrates the induction of longitudinal mechanical oscillation in a metasurface membrane through optical forces. By tuning the radius and separation of the meta-atoms, the membrane can be designed to exhibit either a "repulsive" or "attractive" response. The study presents both experimental data and theoretical calculations to support its findings.

While the physical principles involved are generally well-known, I am inclined to recommend this manuscript for publication. Its novelty lies in the experimental realization of tailored mechanical control over the longitudinal motion of a metasurface.

To the best of my knowledge, previous studies on optically-driven metasurfaces have primarily focused on controlling transverse motion, making this work a significant contribution to the field.

Minor Issues

1. Line 104: It is stated that $F(z) = \alpha \text{Grad } I(z)$, and subsequently mentioned that α is primarily determined by magnetic and electric dipole, quadrupole, and octupole modes. While the authors may be correct, I find this explanation unclear. My understanding is that $F(z) = \alpha \text{Grad } I(z)$ holds for an electric dipole. While this might extend to the flat membrane in question, it is not immediately obvious. Furthermore, I do not see why higher-order multipoles (quadrupole, octupole, etc.) are relevant in this context. Could the authors provide clarification on this point?

2. Potential Applications: Could the authors elaborate on the potential applications of this work? Providing insight into how these findings could be utilized in practical scenarios would enhance the impact of the manuscript.

Reviewer #2

(Remarks to the Author)

In this manuscript, the authors introduce a suspended silicon-based metasurface structure and examine the optical force induced within an optical standing wave field. Employing both numerical simulations and experimental investigations, they demonstrate that varying the structural parameters of the metasurface units enables modulation of the optical force direction (either positive or negative values), thereby facilitating precise control over the sign of optical forces. This work proposes a novel approach for the fine-tuning of optical force manipulation. Nonetheless, it is my assessment that the level of innovation and significance of this work does not yet meet the threshold for publication in Nature Communications, primarily for the following two reasons.

First, some previous researches on optical pulling forces have already revealed that micro- /nano-scale particles can experience both pushing and pulling forces under identical illumination conditions, depending on the particle size and material properties, owing to the Mie scattering mechanism. While this work extends the concept from single particle to periodic disk arrays and demonstrates the optical force reversal by adjusting disk dimensions, it does not reveal fundamentally new effects or underlying physical mechanisms beyond those already known for single particles.

Second, the manuscript lacks a thorough and detailed investigation into the mechanisms of the observed optical forces. Although the authors mention that the reversal of optical force arises from the Mie multipole modes, they do not provide explicit mode decomposition analyses, such as quantifying contributions from electric dipoles, magnetic dipoles, quadrupoles, and higher-order modes. The supplementary information briefly mentions these but remains insufficiently detailed. Furthermore, the study omits analysis of the near-field electromagnetic distributions of the metasurface under conditions of positive and negative force, as well as the corresponding far-field scattering properties. Additionally, the roles of metasurface periodicity and inter-disk coupling in influencing optical forces are not addressed.

The following minor revisions are suggested to enhance the manuscript:

1. A scale bar is needed in Figure 1b to provide readers with a clear understanding of the fabricated metasurface

dimensions.

2. It would be better to discuss whether other polarization states, beyond the y-polarized light used, can also affect the sign or magnitude of the optical force.

3. In line 205, the authors mentioned the use of unstructured thin film vibration induced by single-beam light pressure as the reference for calculating positive or negative forces, despite conducting experiments with a dual-beam standing wave field. It is recommended to justify the appropriateness of this reference choice within the main text.

4. A concise description of the COMSOL simulation conditions should be incorporated into the main text rather than relegated solely to supplementary materials.

5. Figure 4b appears to exhibit a 2π phase discontinuity at the position $z_0/\lambda = -0.2$, which is not addressed in the text. An explanation of this phase anomaly should be provided.

6. Given that the experimental investigations are confined to a wavelength of 1550 nm, it would be beneficial to include numerical simulations predicting whether optical force reversal persists across other spectral regions, particularly within the visible range.

Version 1:

Reviewer comments:

Reviewer #1

(Remarks to the Author)

The authors have satisfactorily addressed all my concerns. As I already stated in previous comments, while the principles involved are known, the other aspects of the manuscript convinced me to recommend it for publications.

Reviewer #2

(Remarks to the Author)

The authors have thoroughly addressed all of my inquiries and have provided pertinent result images accompanied by corresponding discussions. At present, I have only one minor question. In the response, the authors noted that "since Mie resonances are determined by the ratio between the wavelength and the geometrical dimensions, one can access both positive and negative forces simultaneously by using two light fields of different wavelengths." Accordingly, I would suggest that the authors include an example illustrating the effects of varying wavelengths in the supplementary information.

Referee A

The manuscript demonstrates the induction of longitudinal mechanical oscillation in a metasurface membrane through optical forces. By tuning the radius and separation of the meta-atoms, the membrane can be designed to exhibit either a "repulsive" or "attractive" response. The study presents both experimental data and theoretical calculations to support its findings. While the physical principles involved are generally well-known, I am inclined to recommend this manuscript for publication. Its novelty lies in the experimental realization of tailored mechanical control over the longitudinal motion of a metasurface. To the best of my knowledge, previous studies on optically-driven metasurfaces have primarily focused on controlling transverse motion, making this work a significant contribution to the field.

Answer

We thank the reviewer for their positive feedback on the novelty and importance of our manuscript and recommendation for publication.

Minor Issues

1. Line 104: It is stated that $F(z) = \alpha \nabla I(z)$, and subsequently mentioned that α is primarily determined by magnetic and electric dipole, quadrupole, and octopole modes. While the authors may be correct, I find this explanation unclear. My understanding is that $F(z) = \alpha \nabla I(z)$ holds for an electric dipole. While this might extend to the flat membrane in question, it is not immediately obvious. Furthermore, I do not see why higher-order multipoles (quadrupole, octupole, etc.) are relevant in this context. Could the authors provide clarification on this point?

Answer

We agree with the reviewer that this point is not intuitive and needs a more extended discussion. It is correct that $F_{\text{grad}} = \alpha \nabla I(z)$ holds generally only for the electric dipole. Nevertheless, for the special case of a linearly polarized standing wave in the paraxial approximation, one can rewrite the gradient force experienced by the membrane in a similar form $F_{\text{grad}} \approx \alpha_{\text{eff}} \nabla I(z)$ where α_{eff} is the membrane effective polarizability.

Higher-order multipoles are relevant in this context because the Mie coefficients a_l, b_l vary in a complex fashion over the radius range $R = [250\text{nm}, 530\text{nm}]$ such that any order can dominate over the others for a given radius (see Fig. S3a). The Mie coefficients in turn are related to the gradient force through the polarizabilities as e.g. for the electric and magnetic dipole ($l = 1$)

$$\alpha_d^{(e)} = \frac{6\pi i}{k^3} a_1 \quad \alpha_d^{(m)} = \frac{6\pi i}{k^3} b_1. \quad (1)$$

This has the consequence that e.g. for $R = 500\text{nm}$ the Mie coefficient b_3 associated to the magnetic octupole plays an important role.

To clarify this aspect in the manuscript, we added a section to the supplementary material (Sec. II Multipole decomposition of the gradient force in a standing wave) explaining how we approximate the gradient force from each multipole as the product of the intensity gradient and a corresponding multipole polarizability, whose sign alternates with increasing order (due to the involvement of higher-order derivatives).

Action taken: In the main manuscript we highlight this aspect by adding the following sentence

[...] This is intuitive as, for a linearly polarized field $\mathbf{E} \propto \hat{\mathbf{x}}E_0 \cos(kz)$, higher-order multipoles correspond to higher order derivatives, which change sign (see Supplementary Information Sec. II). [...]

2. Potential Applications: Could the authors elaborate on the potential applications of this work? Providing insight into how these findings could be utilized in practical scenarios would enhance the impact of the manuscript.

Answer

We appreciate the opportunity to discuss the potential applications of this work in more depth. Overall, we envision two primary categories of benefits.

The first category extends more globally to optical trapping and manipulation. For instance, trapping a meta-atom at a minimum of intensity could significantly minimize photo-damage. We also anticipate that the use of negative polarizability will aid trapping near surfaces, a regime that is typically challenging to access. The second category relates more specifically to optomechanics, enabling precise engineering of optical forces and thus improved motional control of optomechanical systems. Furthermore, achieving negative polarizability is key, as it can reduce limiting factors such as scattering-photon recoil and light absorption.

More in detail we foresee our work could benefit applications in:

1) **Biological systems** because photodamage in biological samples can be challenging, positioning biospecimen near meta-atoms trapped in lower intensity fields (see also [1]) would open new opportunities.

2) **Particle sorting and assembly** where size dependent force amplitudes and directions enable a size dependent manipulation of particles in space allowing to either purify samples or wavelength dependent placement of "nanobricks" of a certain size in nanofabrication techniques.

3) **Optical trapping:** The ability to tune the sign and magnitude of optical forces by either tuning the geometrical parameters or the wavelength allows the combination of attractive and repulsive forces, creating potentials that allow for controlled and tunable distances, analogue to atoms [2].

4) **Light-driven actuators:** The combination of light based lateral and longitudi-

nal force engineering enables possible 3D steering of meta-vehicles based on engineered metasurfaces.

To develop on potential application of our work, we have added to the conclusion additional applications to steering of metavehicles for drug delivery [3, 4, 5] and optically reconfigurable photonic devices [6].

Action taken: The revised text reads now as follows:

[...] The experimental demonstration of multipolar optical force control holds relevance for diverse applications, including the minimization of photo-damage in biological systems [61], resonant nanoparticle sorting [62], optical manipulation [38] for precise nanostructure assembly, steering of metavehicles for drug delivery [34–36], optically reconfigurable photonic devices [55], near-surface trapping [42], and optomechanics in general. [...]

Referee B

In this manuscript, the authors introduce a suspended silicon-based metasurface structure and examine the optical force induced within an optical standing wave field. Employing both numerical simulations and experimental investigations, they demonstrate that varying the structural parameters of the metasurface units enables modulation of the optical force direction (either positive or negative values), thereby facilitating precise control over the sign of optical forces. This work proposes a novel approach for the fine-tuning of optical force manipulation. Nonetheless, it is my assessment that the level of innovation and significance of this work does not yet meet the threshold for publication in Nature Communications, primarily for the following two reasons.

Answer

We thank the reviewer for recognizing our work as a novel approach to fine-tuning optical force manipulation. Regarding concerns about innovation and significance, we respectfully emphasize that the intrinsic tunability of attraction toward the intensity minimum or maximum — an analogue to atomic systems — has not been previously addressed or realized experimentally.

Earlier studies demonstrated steady-state attraction to the intensity minimum, but relied on non-tunable strategies such as using materials with a refractive index lower than the surrounding medium [7, 8, 9] or employing complex beam-shaping techniques [10, 11, 12, 13]. More recently, nanofabrication has indeed enabled the control of lateral and transverse optical forces [3, 4, 5]; however, tunable transverse forces, analogous to those in atomic systems, remain an open challenge.

We therefore believe that our work provides a significant step forward in this direction.

I. First, some previous researches on optical pulling forces have already revealed that micro-/nano-scale particles can experience both pushing and pulling forces under identical illumination conditions, depending on the particle size and material properties, owing to the Mie scattering mechanism. While this work extends the concept from single particle to periodic disk arrays and demonstrates the optical force reversal by adjusting disk dimensions, it does not reveal fundamentally new effects or underlying physical mechanisms beyond those already known for single particles.

Answer

We thank the reviewer for this comment. We agree with the reviewer that optical forces in general – and negative pulling forces at the nano- and microscale in particular – have been a very active field of research over the last decades, and that interest still persists. Positive optical forces have been enhanced with light-field engineering [14] and

Mie resonances [15, 16]. Nevertheless, we are unable to identify conclusive experimental proof for negative optical pulling forces on dielectric particles due to Mie-resonance engineering. In the work by Monteiro et al. [17], only hints of a sign change of the force on dielectric particles have been reported. Similar in the work of Vcivzmar et al.[18], the authors suggest, based on theoretical analysis, that the bead is trapped at the node of the standing wave. However, both studies do not provide additional experimental verification.

Optical pulling forces have been realized experimentally by engineering the light field [10, 19] and the refractive index [20, 21]. However, the potential for achieving negative forces via Mie-resonance engineering has remained largely unexplored experimentally, and has so far only been studied theoretically for bright trapping (positive forces) [22] and dark trapping (negative forces) [1].

Tunability of optical forces has been achieved using active optical elements such as spatial light modulators [14] and temperature-dependent refractive indices [21]. Moreover, Mie resonances in metamaterials have been exploited for lateral force engineering in two-dimensional planes [3, 4, 5]. However, tunable, longitudinal force engineering based on Mie resonances has, to the best of our knowledge, not yet been experimentally demonstrated. Our experimental work addresses this previously unexplored regime of optical force engineering [1].

Using a suspended, periodic metasurface, where the net optical force results from the geometry-dependent excitation of multipolar modes across the array, we can modulate both the magnitude and sign of the optical force solely by tuning the meta-atom dimensions, without requiring complex structured light fields or exotic material properties. Furthermore, since Mie resonances are determined by the ratio between the employed wavelength and the geometrical dimensions, one can access both positive and negative forces simultaneously by using two light fields of different wavelengths, a feature that is intrinsic to our approach [1].

Action taken: To further highlight the prior work on these related topics, we added the references [19, 22, 17, 18] to the main manuscript. Furthermore we adapted the following sentence from

[...] More recently, drawing an analogy to two-level atoms [42], we proposed to exploit electromagnetic Mie resonances in high-permittivity meta-atoms for optical trapping at intensity minima [43], by exploiting engineered multipolar resonances [44]. [...]

to

[...] More recently, drawing an analogy to two-level atoms [42], we proposed to exploit electromagnetic Mie resonances in high-permittivity meta-atoms for optical trapping at intensity minima [43], by exploiting engineered multipolar resonances [44,45], an effect that has been theoretically anticipated in earlier works [46,47]. [...]

II. Second, the manuscript lacks a thorough and detailed investigation into the mechanisms of the observed optical forces. Although the authors mention that the reversal of optical force arises from the Mie multipole modes, they do not provide explicit mode decomposition analyses, such as quantifying contributions from electric dipoles, magnetic dipoles, quadrupoles, and higher-order modes. The supplementary information briefly mentions these but remains insufficiently detailed. Furthermore, the study omits analysis of the near-field electromagnetic distributions of the metasurface under conditions of positive and negative force, as well as the corresponding far-field scattering properties. Additionally, the roles of metasurface periodicity and inter-disk coupling in influencing optical forces are not addressed.

Answer

We thank the reviewer for raising these points. We address their requests in the following three sections: Multimode decomposition, near field and far field scattering, and influence of periodicity and inter-disc coupling.

Multimode decomposition: To clarify this aspect of mode decomposition, we derive the intensity dependence of the gradient force due to the individual electrical and magnetic dipole, quadrupole and octupole. We find that all considered multipoles scale with $\propto \nabla I$ and that magnetic and electric multipoles of the same order differ in sign. Further more, we find that with each higher order changes the signs. This motivates our approximation of the effective polarizability as

$$\alpha_{\text{eff}} \approx [\alpha_d^{(m)} - \alpha_d^{(e)}] - [\alpha_q^{(m)} - \alpha_q^{(e)}] + [\alpha_o^{(m)} - \alpha_o^{(e)}]$$

where $\alpha_d^{(m)}$ and $\alpha_d^{(e)}$ correspond to the magnetic and electric dipole contributions, $\alpha_q^{(m)}$ and $\alpha_q^{(e)}$ to the magnetic and electric quadrupoles, and $\alpha_o^{(m)}$ and $\alpha_o^{(e)}$ to the magnetic and electric octupole.

Action taken: We added this derivation to the supplementary (Sec. II Multipole decomposition of the gradient force in a standing wave) and added the following sentence to the manuscript.

[...] This is intuitive because higher-order multipoles correspond to higher order derivatives, which change sign for a linearly polarized field $\mathbf{E} \propto \hat{\mathbf{x}}E_0 \cos(kz)$ at each order (see Supplementary Information Sec. II).[...]

Near field and far field scattering: The scattering cross section is directly related to the multimode polarizabilities that are related to the Mie coefficients via $\alpha_l^{(e)} \propto a_l$ and $\alpha_l^{(m)} \propto b_l$. To gain a deeper understanding of the near-field distributions and the individual contributions of different multipoles, Fig. R1a shows the evolution of the Mie scattering coefficients a_l (electric) and b_l (magnetic) as a function of the disk radius R . To make their role in the optical force explicit, in Fig. R1b we regroup the multipole contributions by order: $(b_1 - a_1)$ quantifies the dipole channel, $(a_2 - b_2)$ the quadrupole

channel, and $(b_3 - a_3)$ the octupole channel. This representation highlights the direct competition between electric and magnetic resonances of the same order. For instance, at $R \approx 300$ nm the dipolar term $(b_1 - a_1)$ dominates, yielding a positive force. The corresponding electrical and magnetic near-field profiles for $R = 300$ nm are shown in Fig. R2a and d, respectively, indicating a mix of electric and magnetic dipoles. While at larger radii, for example at $R = 350$ nm, the quadrupolar channel $(a_2 - b_2)$ overtakes, reversing the sign of $\text{Re}(\alpha_{\text{eff}})$ and thus the optical force. The associated electrical and magnetic near-field profiles are shown in Fig. R2b and e, respectively. Additionally, at $R = 500$ nm, the octupole channel $(b_3 - a_3)$ is dominating (see Fig. R2c and f for near-field mode profile) with negative polarizability and giving rise to negative force.

Figure R1: **Multipole polarizabilities and optical forces** a) Mie scattering coefficients $\text{Re}(a_l, b_l)$ as a function of radii R and separation $S = 730$ nm. b) Multipole contribution grouped by dipole channel $b_1 - a_1$ (blue solid line), quadrupole channel $a_2 - b_2$ (red solid line) and octupole channel $b_3 - a_3$ (red dashed line). c) Comparison between the effective polarizability $\text{Re}(\alpha_{\text{eff}})$ (red solid line) and normalized optical forces $F_0/(F_{RP}\mathcal{F})$ (blue solid line) as a function of radii R and separation $S = 730$ nm.

Figure R2: **Near-field electric and magnetic distributions.** Normalized electric (a, b, c) and magnetic (d, e, f) field profiles for disk radii of $R = 300, 350,$ and 500 nm, respectively. The fields are normalized to the background standing-wave amplitude $|E_0|$ and $|H_0|$.

Fig. R1c compares the real part of the effective polarizability $\text{Re}(\alpha_{\text{eff}})$ obtained from this decomposition with the normalized radiation pressure force $F_0/[F_{\text{RP}}\mathcal{F}]$. The close agreement confirms that the alternation between positive and negative forces arises from the interplay of dipolar, quadrupolar, and higher-order multipoles. This explicit decomposition therefore provides a clear physical mechanism for the observed force reversal.

Finally, we depict the simulated far-field scattering for the case of positive ($R = 300\text{nm}$) and negative force ($R = 350\text{nm}$) in Fig. R3. The magnitude of the electrical field along the optical standing wave is shown in Fig. R3a, exhibiting the sinusoidal intensity profile. In the case of positive force for $R = 300\text{nm}$ we observe that the light is mainly backscattered (see Fig. R3b) as we would expect from momentum conservation. In contrast, for the negative force case, the light is mainly scattered in the forward direction (see Fig. R3c) pulling the metastructure towards $z < 0$.

Figure R3: **Directional scattering of meta-atoms.** a) Electrical field $|E_0|$ of the optical standing wave with the positive intensity slope at $z = 0$. b) Positive force towards $z > 0$ for $R = 300\text{nm}$ and $S = 730\text{nm}$. c) Negative force towards $z < 0$ for $R = 350\text{nm}$ and $S = 730\text{nm}$.

Figure R4: **Multipolar decomposition of the scattering cross section.** The scattering cross section is shown as a function of disk radius R for a fixed separation $S = 730$ nm. The contributions from different multipole channels are indicated as follows: electric dipole (ED, blue solid line), electric quadrupole (EQ, blue dashed line), magnetic dipole (MD, red solid line), magnetic quadrupole (MQ, red dashed line), and magnetic octupole (MO, red dash-dotted line). The evolution of these contributions highlights the transition between dominant multipolar modes with increasing disk radius.

To investigate the scattering behavior and the contribution of individual multipolar channels, Fig. R4 shows the decomposition of the scattering cross section into the contributions of each multipole. We fix the separation to $S = 730$ nm where we obtained the most data points exhibiting negative forces. The features closely mirror those observed for the multipole polarizabilities and their influence on the optical force F_0 (see Fig. R1). For example, at $R = 300$ nm, the scattering is dominated by the electric dipole ED contribution, whereas at $R = 350$ nm, the electric quadrupole EQ becomes predominant. At larger radii, such as $R = 500$ nm, the magnetic octupole MO is the principal contributor to the scattering cross section.

Action taken: We adapted Fig. R1a by adding the electrical octupole and added Fig. R2 and R3 and their discussion to the supplementary section III.

Influence of periodicity and inter-disc coupling To address the reviewer's remark we turn to Fig. 3 of the main text where we scan the separation S . Generally, the force landscape is less sensitive to the separation than to the radius. But we can still distinguish different regimes. For very small and very large radii such as $R < 300$ nm and $R > 400$ nm, we see negligible influence of the separation on the force sign and

magnitude. In contrast, for $300\text{nm} < R < 400\text{nm}$ we observe a reversal of the force direction with the separation. We attribute this to the fact that for very small radii ($R \ll \lambda$) the scattered field leaks out of the disk into the connection beam, creating a significant and stable contribution to the scattered field, where higher multipoles from the disc can be neglected. In contrast, for very large radii ($\lambda/2 < R$) the scattered field is mostly concentrated in the disc itself, and the connection beam can be neglected. In the intermediate regime, ($R \simeq \lambda/3$) we observe a transition between the two regimes, leading to some influence of the separation on the force landscape.

Action taken: We clarify this point in the manuscript by changing the following sentence from

[...] We observe a broad tunability of the force F_0 that is only weakly affected by the separation S . [...]

to

[...] We observe a broad tunability of the force F_0 that is only weakly affected by the separation S , with its strongest influence occurring in the intermediate regime ($R \simeq \lambda/3$). We attribute this to the scattered field stemming from the connecting beams near the disc and the disc itself.

[...]

The following minor revisions are suggested to enhance the manuscript:

1. A scale bar is needed in Figure 1b to provide readers with a clear understanding of the fabricated metasurface dimensions.

Answer

We thank the reviewer for this helpful suggestion. Action taken: We have revised Figure 1b and included a scale bar, which clarifies the dimensions of the fabricated metasurface.

2. It would be better to discuss whether other polarization states, beyond the y -polarized light used, can also affect the sign or magnitude of the optical force.

Answer

We thank the reviewer for this valuable suggestion. The metasurface unit cell is symmetric in the lateral dimensions (the xy -plane). Consequently, the structure responds identically to any linear polarization, and the resulting optical force along the z -direction has the same magnitude and sign for both x - and y -polarized light. To illustrate this, in Fig. R5 we compare the z -component of the optical force, F_0 , under x -polarized, y -polarized and right-handed circularly polarized (RHCP) excitations. As shown, F_0

remains identical in both magnitude and sign. However, in the case of RHCP, a torque component τ_z emerges, as presented in Fig. R6. Interestingly, this torque can also be tuned from positive to negative values. Such tunability not only highlights the versatility of our design but also opens up promising opportunities for applications in optomechanical, light-driven rotation, and advanced concepts such as meta-vehicles but reach beyond the scope of our manuscript.

Action taken: To address the reviewer’s suggestion we added the following sentence to the manuscript:

[...] Note that force amplitude and direction is polarization independent due to the symmetry of the meta-atom array (see Supplementary Information Sec. III). [...]

Furthermore, we have added Fig. R5 and Fig. R6 to the Supplementary Information (Sec. III) to provide additional information and visualization of the polarization-independent optical force and torque under illumination with circular polarized light.

Figure R5: **Polarization independence of optical force.** The optical force F_0 for different radius R under linear polarization along x (solid blue), y (dashed red) and right-hand circular polarization RHCP (dotted black).

Figure R6: **Emerging torque under circular polarization.** Torque τ_z for different radius R under right-hand circular polarization (RHCP). The torque assumes positive and negative values.

3. In line 205, the authors mentioned the use of unstructured thin film vibration induced by single-beam light pressure as the reference for calculating positive or negative forces, despite conducting experiments with a dual-beam standing wave field. It is recommended to justify the appropriateness of this reference choice within the main text.

Answer

We thank the reviewer for this comment. The unstructured thin film under single-beam illumination is used as a reference because the radiation pressure in this configuration can be calculated analytically and simulated numerically, providing a well-defined normalization for the optical forces. Furthermore, using single-beam radiation pressure allows us to verify the experimental setup, as the force is always pushing along the beam propagation axis, and the resulting phase response of the membrane motion is deterministic. This reference enables a consistent and reliable assignment of positive and negative forces when measuring the response in the dual-beam standing-wave configuration.

Action taken: To clarify this aspect, we added the following sentence to the manuscript:

[...] Importantly, the single beam reference measurement allows a reliable calibration of the phase $\Theta(\omega)$ due to the well-defined force direction along the beam propagation axis, for an unstructured flat membrane under single beam illumination. [...]

4. A concise description of the COMSOL simulation conditions should be incorporated into the main text rather than relegated solely to supplementary materials.

Answer

We thank the reviewer for this helpful suggestion. Action taken: Following the recommendation, a concise summary of the COMSOL simulation setup has been removed from the method section and added to the main text, while the complete description remains in the Supplementary Information (Section S1).

[...] COMSOL Multiphysics simulation in a scattered-field formulation models a single silicon meta-atom suspended in air within a periodic unit cell. Periodic boundary conditions (PBC) are applied laterally, and perfectly matched layers (PML) along the propagation axis to suppress reflections. A normally incident standing wave provide the excitation, and the optical forces are extracted via Maxwell stress tensor methods [23] using the total electromagnetic field (see Supplementary Sec. III). [...]

5. Figure 4b appears to exhibit a 2π phase discontinuity at the position $z_0/\lambda = -0.2$, which is not addressed in the text. An explanation of this phase anomaly should be provided.

Answer

We thank the reviewer for pointing this out. The apparent 2π phase discontinuity in Figure 4b arises from phase wrapping in the lock-in amplifier. The lock-in measures the phase modulo 2π , so when the phase crosses the $\pm\pi$ boundary, it appears as a sudden jump of 2π in the plotted data. The underlying physical phase of the membrane motion varies smoothly; the discontinuity is purely a measurement artifact.

Action taken: We have clarified this point by adding the following sentence to the main text:

[...] The phase discontinuity observed at $z_0/\lambda = -0.2$ arises from phase wrapping in the lock-in detection (limited to $\pm\pi$) which implies a sudden phase jump by 2π when $\Theta(\omega_{\text{dr}})$ crosses $\pm\pi$.

6. Given that the experimental investigations are confined to a wavelength of 1550 nm, it would be beneficial to include numerical simulations predicting whether optical force reversal persists across other spectral regions, particularly within the visible range.

Answer

We thank the reviewer for this valuable suggestion. To address it, we performed additional simulations at the common visible wavelength of 632 nm, while keeping the same exemplary periodicity ($S = 730$ nm) and sweeping the disk radius from 250 nm to 550 nm.

Action taken: The results are presented in Fig. R7 and added to the Supplementary Information Sec. III.

Since the lattice separation S is larger than the wavelength λ , the meta-atoms behave more like isolated scatterers. Consequently, the inter-disk coupling is relatively weak in this regime which results in reduced optical forces F_0 compared to the near-infrared case. Nevertheless, an important observation is that the *sign* of the optical force can still be controlled by tuning the geometrical parameters of the meta-atoms.

While the overall magnitude of the force is weaker due to reduced coupling, we believe the force can still be significantly enhanced through careful optimization of the geometric parameters, particularly the separation S and radius R . These findings confirm that the principle of optical force reversal is not limited to the telecom wavelength range but can also be extended to the visible spectrum. This highlights both the robustness of the concept and its potential applicability across a broad spectral range.

Figure R7: Transversal optical force F_0 versus radius R for separation $S = 730\text{nm}$ at wavelength $\lambda = 633\text{nm}$.

References

- [1] S. Lepeshov, N. Meyer, P. Maurer, O. Romero-Isart, and R. Quidant, *Phys. Rev. Lett.* **130**, 233601 (2023).
- [2] E. Vetsch, D. Reitz, G. Sagué, R. Schmidt, S. Dawkins, and A. Rauschenbeutel, *Physical Review Letters* **104**, 203603 (2010).
- [3] D. Andrén, D. G. Baranov, S. Jones, G. Volpe, R. Verre, and M. Käll, *Nature Nanotechnology* **16**, 970 (2021).
- [4] T. Li, J. J. Kingsley-Smith, Y. Hu, X. Xu, S. Yan, S. Wang, B. Yao, Z. Wang, and S. Zhu, *Optics Letters* **48**, 255 (2023).
- [5] E. Engay, M. Shanei, V. Mylnikov, G. Wang, P. Johansson, G. Volpe, and M. Käll, *Light: Science & Applications* **14**, 38 (2025).
- [6] A. Afridi, J. Gieseler, N. Meyer, and R. Quidant, *Nano Lett.* **23**, 2496 (2023).
- [7] K. Gahagan and G. Swartzlander Jr, *Optics Letters* **21**, 827 (1996).
- [8] K. Gahagan and G. Swartzlander Jr, *Journal of the Optical Society of America B* **15**, 524 (1998).
- [9] M. P. MacDonald, L. Paterson, W. Sibbett, K. Dholakia, and P. E. Bryant, *Optics Letters* **26**, 863 (2001).
- [10] J. Chen, J. Ng, Z. Lin, and C. T. Chan, *Nat. Photonics* **5**, 531 (2011).
- [11] W. Ding, T. Zhu, L.-M. Zhou, and C.-W. Qiu, *Adv. Photonics* **1**, 024001 (2019).
- [12] V. Shvedov, A. R. Davoyan, C. Hnatovsky, N. Engheta, and W. Krolikowski, *Nat. Photon.* **8**, 846 (2014).
- [13] O. Brzobohatý, V. Karásek, M. Šiler, L. Chvátal, T. Čižmár, and P. Zemánek, *Nature Photonics* **7**, 123 (2013).

- [14] M. A. Taylor, M. Waleed, A. B. Stilgoe, H. Rubinsztein-Dunlop, and W. P. Bowen, *Nature Photonics* **9**, 669 (2015).
- [15] A. Andres-Arroyo, B. Gupta, F. Wang, J. J. Gooding, and P. J. Reece, *Nano Lett.* **16**, 1903 (2016).
- [16] N. O. Länk, P. Johansson, and M. Käll, *Opt. Express* **26**, 29074 (2018).
- [17] T. Monteiro, J. Millen, G. Pender, F. Marquardt, D. Chang, and P. Barker, *New Journal of Physics* **15**, 015001 (2013).
- [18] T. Čižmár, V. Garcés-Chávez, K. Dholakia, and P. Zemánek, *Appl. Phys. Lett.* **86**, 174101 (2005).
- [19] A. Dogariu, S. Sukhov, and J. Sáenz, *Nature Photonics* **7**, 24 (2013).
- [20] P. Prentice, M. P. Macdonald, T. Frank, A. Cuschieri, G. Spalding, W. Sibbett, P. Campbell, and K. Dholakia, *Optics Express* **12**, 593 (2004).
- [21] L. Mao, I. Toftul, S. Balendhran, M. Taha, Y. Kivshar, and S. Kruk, *Laser Photonics Rev.* **19**, 2400767 (2024).
- [22] A. B. Stilgoe, T. A. Nieminen, G. Knöner, N. R. Heckenberg, and H. Rubinsztein-Dunlop, *Opt. Express* **16**, 15039 (2008).
- [23] J. D. Jackson, *Classical electrodynamics* (John Wiley & Sons, 2021).

Referee A

The authors have satisfactorily addressed all my concerns. As I already stated in previous comments, while the principles involved are known, the other aspects of the manuscript convinced me to recommend it for publications.

Answer

We thank the reviewer for their recommendation for publication.

Referee B

The authors have thoroughly addressed all of my inquiries and have provided pertinent result images accompanied by corresponding discussions. At present, I have only one minor question. In the response, the authors noted that "since Mie resonances are determined by the ratio between the wavelength and the geometrical dimensions, one can access both positive and negative forces simultaneously by using two light fields of different wavelengths." Accordingly, I would suggest that the authors include an example illustrating the effects of varying wavelengths in the supplementary information.

Answer

To fulfill the reviewer's request we added another section named *Wavelength dependence of the optical force* to the supplementary material:

[...]

To exploit the fact that Mie resonances are governed by the ratio between the wavelength and the structural dimensions, one can tune the illumination wavelength to access both positive and negative optical forces for a fixed metasurface design. Fig. R1 illustrates this principle by showing the normalized optical force $F_0/(F_{RP}\mathcal{F})$ as a function of wavelength for the representative M_- structure under x -polarized illumination. The wavelength

Figure R1: **Wavelength dependence of the optical force.** The normalized optical force $F_0/(F_{RP}\mathcal{F})$ as a function of the wavelength λ for the M_- structure under linearly polarized (x -polarized) illumination.

sweep reveals a pronounced tunability of both the magnitude and the sign of the optical force, demonstrating simple spectral control between attractive and repulsive forces. [...]